# Impact, obstacles and boundaries of patient partnership: A qualitative interventional study in Lebanon

**Alaa Yehya Dayekh**[1,2]*, **Mohammad Naseridine**[1,3]*, **Fatima Dakroub**[1], **Adel Olleik**[3]

**1** Faculty of sciences, Lebanese University, Beirut, Lebanon, **2** Quality Improvement Department, Saint George Hospital, Beirut, Lebanon, **3** Gates Group Society sarl., Beirut, Lebanon

* alaadayekh@hotmail.com (AYD); Mohamad.nassereddine@ul.edu.lb (MN)

**Data Availability Statement:** All relevant data are available on Fighare: https://figshare.com/articles/dataset/Dayekh_et_al_Patient_Partnership_2022_xlsx/19611651.

## Abstract

The patient as partner approach is a modern model of patient engagement that integrates the patients' knowledge and skills into managing their own health. This study aims to evaluate the benefits and barriers of patient partnership in a healthcare setting. It is a qualitative and interventional study that implemented a patient and family partnership committee (PFPC) at a Lebanese hospital during the COVID-19 pandemic. A purposeful guided approach was used for sampling, and data was collected by structured questionnaire interviews. Seven PFPC team dynamics building blocks were generated: transparency, support, motivation, comfortable communication, mutual understanding, equity in positions and empowerment to participate. Both the patient partners (94%) and healthcare professionals (90%) were satisfied with the PFPC experience. The majority of the healthcare professionals (HP) reported a noticeable change in the quality improvement process (QIP) (89%) and approved to standardize the PFPC (93%). The patient partnership benefits were clear, and the PFPC was perceived positively by both patient partners (PP) and HP. PP experienced distress relief (37%), gained ideas (41%) and felt that their opinion was heard (27%) after PFPC participation. PP reported benefits to hospitalized patients, including respect and care (63%) and patient satisfaction (20%). The main challenges for PFPC implementation were time availability and conflicts. Lessons from patient partnership can be utilized to improve the patient care policies in the Lebanese healthcare system. Moreover, developing countries can benefit from the patient partnership approach in their healthcare settings.

## Introduction

Quality is defined as the patient's overall judgment on the excellence of healthcare services [1]. Although the quality of care is linked directly to patients, they are excluded during the quality improvement process (QIP). In this paternalistic model of care, patients are not involved in important QIP such as areas for improvement analysis and decision-making. The World Alliance for Patient Safety urges health care organizations to direct their healthcare improvement efforts into patient and family participation [2]. The patient as partner concept integrates the involvement, participation, engagement and empowerment terms. These terms are

**Funding:** The author(s) received no specific funding for this work.

**Competing interests:** The authors have declared that no competing interests exist.

**Abbreviations:** CEPPP, Center of Excellence on Partnership with Patients and the Public; HP, Healthcare Professionals; PFCC, Patient and Family Centered Care; PFPC, Patient and Family Partnership Committee; PP, Patient Partners; QI, Quality Improvement; QIP, Quality Improvement Process; SGH, Saint George's Hospital.

differentiated by the extent of the patient intervention. The highest rank of the patient and caregiver relationship ladder is patient partnership. The intermediate one is patient participation, in which the healthcare staff assesses a patient's knowledge and provides education accordingly. The lowest rank is the patient involvement in concerns such as the timing of visits and daily living recommendations [3]. Patient partnership represents the optimal mean of patient participation in healthcare quality improvement. Several studies have proposed different definitions for patient partnership, which are summarized in Table 1.

The first patient partnership office was established in Canada by a chronically ill patient who believed in the experiential knowledge acquired during a long illness [8]. A study in Montreal revealed that trained patients who participated in quality improvement committees suggested solutions that were more simple to implement than those proposed by professionals [9]. Another study exploring patient partnership in Florida focused on the shared decision-making between patients, their families and the diagnostic team. It revealed the importance of information disseminated by patients in improving the clinical care delivered by practitioners [10]. Patient partnership also supports professional development by valuing the voice of the front-line workers, fostering a trust culture and providing problem-solving authority to employees [11]. Empowerment of staff in decision-making results in sustainable improvements of the healthcare system.

Patient partners and healthcare providers identified many barriers to patient partnership. The most challenging one was accepting the new healthcare giver-role of patients [12]. Additional barriers were the lack of health literacy [12], demographic variables, disease severity [12], lack of computer and internet access [13], time availability and mental health [14]. Moreover, healthcare providers tend to question the knowledge of individuals without medical training or academic degrees [15].

Patient partnership initiatives are centralized in developed countries, especially in Canada's center of excellence on partnership with patients and the public (CEPPP) [8,9]. Still, the boundaries of patient partnership have not been assessed yet. There's little to no evidence of research on partnership in developing countries. The Lebanese accreditation system allocated a whole chapter for the patient and family rights as a base for engagement [16], but failed to mention patient partnership. This study aims to explore the benefits and barriers of implementing a patient and family partnership committee (PFPC) in an under-resourced and low-

**Table 1. The different patient partnership definitions from literature.**

| Source | Patient partnership definition |
|---|---|
| Fumagalli Health Policy [4] | The concept of patient engagement integrates the patient's knowledge, skills, individual ability and willingness to actively participate in managing his own health with the interventions provided by the healthcare practitioners, designed to increase activation and promote positive patient behavior. |
| Vahdat Iranian Red Crescent Medical Journal [5] | The engagement of the patient in decision making by expressing opinions and thoughts about different treatment choices, which includes a coordinated sharing of information and feelings between the patient and the healthcare professionals while accepting health team instructions. |
| The African Partnerships for patient safety (APPS) [6] | A collaborative relationship between two or more parties based on trust, equality and mutual understanding for the achievement of a specified goal. Partnerships involve risks as well as benefits, making shared accountability critical. |
| The Centers for Medicare and Medicaid Services (CMS) [7] | The patients and families are partners and are crucial in defining, designing, participating in and assessing the care processes and systems that deliver care to them in a respectful manner and based upon patient preferences, needs, and values. |

income country. Moreover, it aims to determine the perception of participants on the extent of patient contribution to the QIP through the PFPC. This study also aims to assess the decision-making boundaries of the patient partners in PFPC. Our results will create knowledge and tools to initiate, implement and evaluate patient partnership in healthcare settings in Lebanon.

## Materials and methods

### Research design

This qualitative study included 68 participants from Saint George's hospital (SGH) between July 1st, 2020 and September 15th, 2020. We conducted descriptive analysis on data collected from 27 healthcare professionals (HP) and 41 patient partners (PP).

### Ethical consideration

The Saint George's Hospital ethical committee approved this study. Written consents were obtained from the PFPC members who attended the direct meetings. Those who attended online provided verbal consents. Data was kept confidential and used for research purposes only.

### Recruitment and selection

A total of 68 participants were included in this study. Before selection, an online video on the value of partnership was disseminated. It was intended to increase awareness and encourage participation among patients, family members and healthcare workers. The sampling strategy was based on a purposeful theory-guided sampling approach [17], which assumes that certain participants with specific characteristics are good candidates to reach the study's objectives. An eligibility screening questionnaire—defined by the inclusion criteria—was implemented in the selection and recruitment of patient partners. It was available in a google survey form and disseminated by the social media platforms of SGH. Since it came back without responses, we directly contacted random potential patients and family members who were staying at the hospital during the recruitment phase. Out of 1997 hospitalized patients, we contacted 70 candidates. Only 41 patients and family members were eligible and willing to participate as a PP. Out of 389 healthcare employees, 121 were multidisciplinary healthcare workers with a minimum experience of two years and previous quality improvement expertise. However, only 27 HP were willing to participate, considered eligible and included.

The inclusion criteria for the PP were as follows: having—personally or through a family member—a wealth of information regarding hospitalization from a chronic illness experience (i), being in a period of stable health to allow safe participation during meetings (ii), having a constructive critical attitude and an ability to identify areas for improvement (iii), possessing good personal communication skills (iv), driven by the will to help others beyond personal experience enhancement (v), being available and willing to allocate time and effort (vi) and being treated at SGH or other healthcare facilities (vii).

Regarding the HP's inclusion criteria, the study enrolled: multidisciplinary healthcare workers with a minimum of two years of experience to facilitate the evaluation of healthcare services (i); hospital leaders, since they are the primary policymakers (ii); physicians with different specialties (iii); healthcare workers with previous quality improvement expertise and training (iv); healthcare workers working in SGH or other healthcare facilities to enrich the quality improvement discussion (v). These assumptions used for successful implementation were carefully studied and compared to similar methodologies from literature [3,8,9].

Participants were excluded if they were less than 18 years old (i), unwilling to participate in the PFPC meetings (ii) or not motivated to enhance the healthcare experience (iii). PP were excluded if they didn't have a chronic illness experience. Finally, HP with less than two years of professional experience were excluded.

## Methodology

The partnership coordinator provided individualized 15-minutes training to each participant before the PFPC meetings. Training topics included the patient partnership concept, QIP and medical and quality terminology. The PFPC was implemented in focus groups consisting of at least 2 PP and 2 HP. Its time length ranged between 60 and 90 minutes. The participants selected a QIP topic to discuss in the PFPC among three options: conversation time with patients (i), workplace violence (ii) and hand hygiene compliance (iii). Participants who attended direct meetings were provided with a free meal and parking space. The process for PFPC implementation is summarized in Fig 1.

## Data collection

We collected data using direct or online individualized interviews. We utilized two different structured qualitative questionnaires for PP and HP. Patients, healthcare providers and research experts checked the questionnaires for their content validity, length appropriateness and questioning flow. The language in both questionnaires was clear and comprehendible. To explore the theoretical framework, many data points should be generated in qualitative research. The questions in both interview guides were open leading to the emergence of specific themes. Interviews were conducted between August 2nd, 2020 and September 5th, 2020. They ranged in length between 10 and 30 minutes. Theoretical saturation was attained

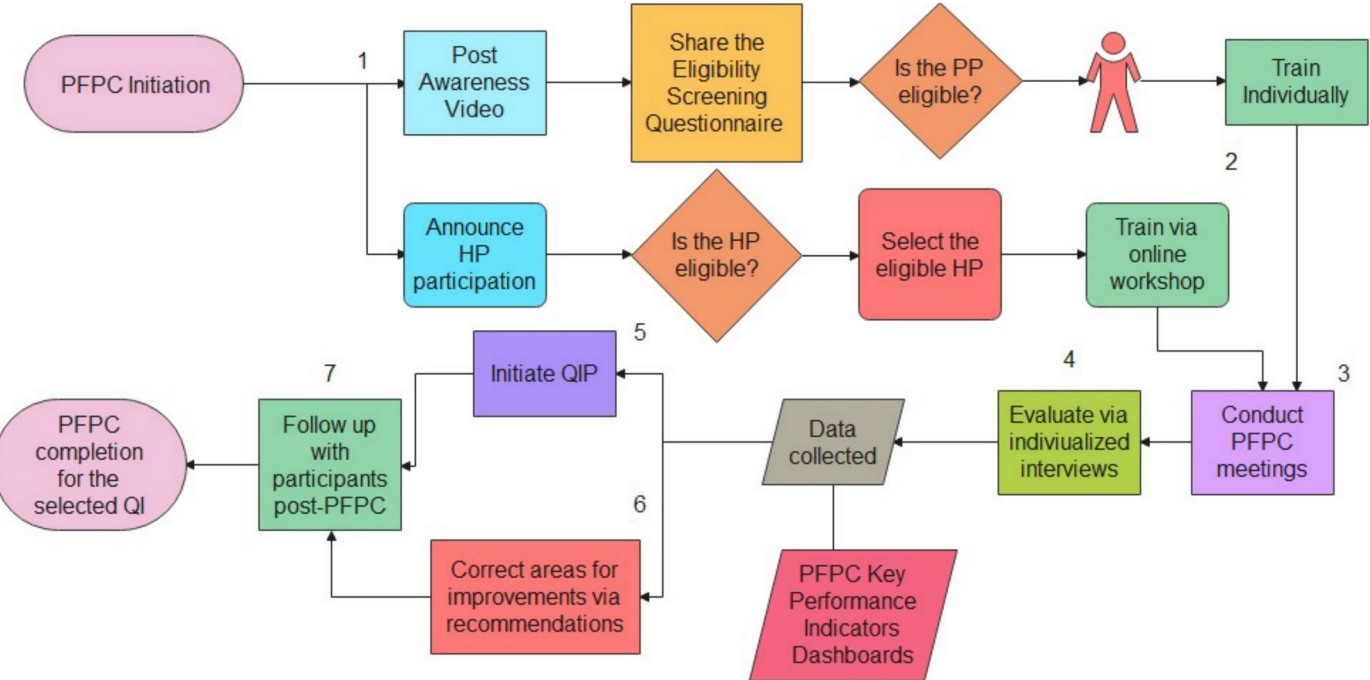

**Fig 1. Flowchart of the patient and family partnership committee process.** (1) PFPC is initiated by posting a video online to increase the awareness of patients on patient partnership. The opportunity for participation in the PFPC is announced for HP at the healthcare setting. (2) After recruiting eligible participants, patient partners are trained individually by the PFPC coordinator. Healthcare professionals are also trained prior to the meetings. (3) The PFPC meetings are conducted directly or online, and (4) participants are evaluated by individualized interviews. (5) After data collection, the quality improvement process can be initiated. (6) Moreover, the suggested opportunities for improvement can be addressed. (7) Participants are followed-up post-PFPC.

following the completion of 15 out of 25 PFPC and 50 out of 68 interviews. Thus, the data collected had identified and filled the properties of the theoretical category [17]. The remaining PFPC and interviews did not produce new ideas, but rather confirmed and stabilized the previous theoretical leads.

## Data analysis

We used the grounded theory approach for data analysis [17]. We analyzed qualitative data from the inductive inquiries in parallel with data collection. We initiated analysis by an open coding phase following the first 10 interviews. Phase one consisted of reviewing the interview transcripts, identifying repeating themes, defining excerpts, and tagging them into an initial coding system. The second phase of analysis was the selective coding, where we organized the primary codes developed and grouped them into categories. A thematic coding structure was hence created by linking, comparing and discussing data. We had utilized the paper coding technique to conduct the different stages of coding strategies. The approved coding categories were then applied to all interview transcripts to ensure grounding of the data. This established a conceptual outline of the participants' perception of the patient partnership approach. The research team constantly interpreted possible relations and compared emergent themes until attaining theoretical saturation. The coded data was entered and analyzed using the IBM SPSS statistics software (*IBM SPSS Statistics for Windows*, *Version 25.0. Armonk*, *NY*: *IBM Corp.*). Descriptive analysis was performed on data collected from interviews with patients and HP. The research team focused on identifying and minimizing potential sources of bias. The sampling method may constitute a potential source of bias by only recruiting patients who were more capable or willing to participate. Another concern was that some participants may have answered the interview questions in a manner which they perceived as socially acceptable. To mitigate this bias, we conducted individualized interviews, provided definitions of concepts with examples as needed and related concepts to scenarios. The subjective nature of qualitative studies complicates a researcher's attempt to completely detach from the data. To minimize bias during data interpretation, we recorded summaries of interviews and conducted regular research meetings. Moreover, two researchers independently analyzed the data and then compared their results jointly.

# Results

## Sample characteristics

Of 68 eligible and included participants, 27 (40%) were HP and 41 (60%) were PP. Half of the PP were represented by family members (51%). The majority of PP occupied jobs that were not related to the healthcare industry (76%). Cancer (66%) was the most common chronic illness among PP. Only 37% of PP had at least 10 years of experience with healthcare services. Nurses took the lead among HP who participated in the PFPC (41%) followed by leaders with managerial titles (33%), quality and safety specialists (22%) and physicians (4%).

Females constituted the majority of PP (61%) and HP (70%) participants (Table 2). The median age of PP and HP was 43 and 28 years respectively. Most PP (71%) and HP (59%) participated directly in the PFPC meetings. All participants chose the QIP topic pertaining to the conversation time with patients.

## The team collaborative process and dynamics

All PP agreed that the PFPC was easy to join online or through direct meetings. PP were active members and felt comfortable communicating their ideas. The majority of HP had an excellent

**Table 2. Sample characteristics of the healthcare professionals and the patient partners.**

| Sample Characteristics | Healthcare professionals (n = 27) | Patient Partners (n = 41) |
|---|---|---|
| **Gender, n (%)** | | |
| Male | 8 (30%) | 16 (39%) |
| Female | 19 (70%) | 25 (61%) |
| **Age, n (%)** | | |
| Less than 40 Years | 24 (89%) | 18 (44%) |
| More than 40 Years | 3 (11%) | 23 (56%) |
| **Age, median (SD)** | 28 (6.79) | 43 (13.57) |
| **Selected QIP topic[1], n (%)** | | |
| Conversation Time With Patients | 27 (100%) | 41 (100%) |
| **Meeting type, n (%)** | | |
| Direct Meeting | 16 (59%) | 29 (71%) |
| Online Meeting | 11 (41%) | 12 (29%) |

[1] Selected from three suggested topics: Conversation time with patients (i), workplace violence (ii), and hand hygiene compliance (iii).

Abbreviations: QIP, Quality Improvement Process; SD, Standard deviation.

impression of the PP solutions (78%). HP agreed that PP were active members (89%) who were comfortable in communicating ideas (93%). HP had to change some quality terminology used in the PFPC meetings.

PP considered having a motivation to participate (44%), ideas (32%) and personal experience (24%) as supporting elements for the PFPC. Moreover, HP expressed the need to have an appropriate PFPC time and place (26%), empowered patients (59%) and transparency (15%).

## The PFPC advantages

Both HP and PP learned from the patient partnership experience and acquired communication and listening skills. Moreover, HP reported gaining brainstorming (11%) and interview skills (19%) and PP reported respect (5%) and analytical skills (12%).

Both PP and HP expressed benefits to hospitalized patients including respect and care, patient satisfaction and enhanced patient experience. Only PP reported the shared decision-making benefit (10%).

Regarding the benefits gained by participants, PP experienced distress relief (37%), felt their opinion was heard (27%), were satisfied by the new experience (12%) and gained ideas (41%). HP reported benefits of mutual understanding (48%), gaining new experience (37%), shared decision making (22%) and work enhancement (4%).

Concerning the PFPC implementation advantages, both PP and HP reported patient loyalty preservation (i), achieving patient satisfaction (ii) and quality improvement without limitations (iii). PP additionally expressed that patients can finally receive their rights (49%). Moreover, HP mentioned PFPC advantages of high-quality care delivery (30%), frontline workers empowerment (22%) and the bolstering of mutual understanding (11%).

## The satisfaction with the partnership approach

The majority of PP (94%) and HP (90%) were highly satisfied with the patient partnership experience. HP reported a noticeable change in the QIP (89%) and approved standardizing the PFPC at SGH (93%).

### The challenges facing PFPC implementation

All HP identified challenges facing the patient partnership process, and around half of PP (54%) did not identify any. The main two challenges mentioned by both PP and HP were conflicts and time availability. Only PP considered the COVID-19 pandemic as a challenge (17%). The challenges that were solely expressed by HP were confidentiality (11%), PP distraction into personal stories (19%), implementation of the PP suggested solutions (37%) and the PP's educational level (26%).

### The opportunities for PFPC improvement

Only 35% of all participants did not recommend any improvements. The main opportunities for improvement were identified as allocating additional PFPC time and providing pre-PFPC training through workshops and hospital rounds. Recommendations that were solely suggested by PP were post-PFPC follow-up (10%), permitting authorized decision-making (10%) and pre-scheduled meetings (7%). Suggestions unique to HP were catering (15%), PFPC policy development (15%) and implementing directed questions instead of open discussions (15%).

### The PFPC coordinator

Both PP (100%) and HP (96%) received the needed support and guidance from the PFPC coordinator. The latter's role was identified by both groups as the support and motivation to participate (i) and the explanation of unclear concepts (ii). HP considered the discussion management as an additional role of the PFPC coordinator.

### The PFPC autonomy boundaries

We assessed the safety of allowing PP to become authorized decision-makers. Participants were asked to choose between healthcare quality concepts and their personal preferences in different decision-making scenarios. Most participants chose concepts directly related to safety and quality over personal preferences (Table 3).

## Discussion

Despite variations in methodology, patient partnership initiatives usually produce favorable outcomes. In this study, we aim to determine the impact and limitations of PFPC implementation in a hospital setting. We assessed the PFPC advantages, barriers for implementation and the opportunities for improvement. Unlike previous studies, we assessed the boundaries of delegating decision-making to the patient partner.

**Table 3. The autonomy boundaries of decision-making compared between healthcare professionals and patient partners.**

| Safety and Quality Concepts | Healthcare Professionals n (%) | Patient Partners n(%) |
| --- | --- | --- |
| Responsibility Toward Other Patients | 25 (93%) | 40 (98%) |
| Medical Recommendations | 18 (67%) | 30 (73%) |
| Resource Utilization | 22 (81%) | 30 (73%) |
| Roles and Regulations | 23 (85%) | 31 (76%) |
| Quality and Safety Opinion | 27 (100%) | 34 (83%) |
| Nurses' Recommendations | 25 (93%) | 38 (93%) |
| Protecting the Common Good | 24 (89%) | 34 (83%) |
| Protection From Danger | 27 (100%) | 39 (95%) |
| Evidence Based practice | 22 (81.5%) | 35 (85%) |

Whether direct or virtual, PFPC meetings were easily accessible for participants in our study. Unlike previous studies [18], pre-training for a long period was not feasible due to the COVID-19 pandemic. Black et al. suggested recommendations to achieve meaningful engagement in successful PFPC meetings [13]. Moreover, certain studies reported the PFPC elements of support solely from the patient partners perspectives [9]. We assessed the engagement factors according to both HP and PP perspectives. We identified seven PFPC team dynamics building blocks: transparency, support, motivation, comfortable communication, mutual understanding, equity in positions and empowerment to participate.

Similar to previous results [9], participants in this study reported a continuous learning experience. Communication and listening skills were improved in both PP and HP. The education provided to HP before PFPC implementation may have been beneficial in increasing their awareness. The skills acquired by front liners through PFPC bolsters the notion that partnership targets the human and talent management dimension.

We found partnership benefits similar to those reported in literature [3,8,9]. A strong point of this study is that it evaluates the benefits gained by hospitalized patients as well as the PFPC participants. The PFPC advantages achieved were not related to the PP being well-educated, young or involved in the healthcare domain. Thus, our eligibility screening questionnaire was a reliable tool for PP recruitment. Despite the objective selection process, HP still tend to doubt the patient's knowledge and thinking [19]. In our study, 26% of HP considered the PP education level as a challenge for PFPC implementation. This emphasizes the importance of pre-PFPC training for both PP and HP.

Participants in this study recommended several opportunities for improvement to enhance the partnership process, which were beyond our research objectives. The recommendations were not implemented due to circumstantial limitations. The COVID-19 pandemic prevented conducting the pre-meeting workshop. Moreover, it reduced the feasibility of holding direct PFPC meetings. Another pandemic-specific limitation was the fear of hospital-acquired infection, which may have affected recruitment. The poor internet connection faced by under-resourced patients may have also restricted participation. Similar to previous studies [13], our project spanned a relatively short time. Moreover, our evaluation of the partnership approach was based on interviews only. Unlike other studies [9], we did not clinically assess the positive outcomes from partnership. Finally, financial support is a crucial element in partnership studies, but it was lacking in ours.

The PFPC coordinator played an important role in achieving participants' satisfaction. To maximize the involvement of partners, he motivated and empowered them and managed discussions in the committee. This role was only mentioned in literature by the Montreal university study under the term 'patient coach' [9]. Our study emphasizes that the professional and supportive communication of a partnership coordinator is crucial for partnership success.

We assessed for the first time the boundaries of delegating the decision-making power to patient partners. Participants were asked to choose between quality and safety concepts in healthcare and their personal preferences. The PP participants demonstrated responsibility and altruism by adhering to quality and safety concepts. Thus, the PP recruited by objective eligibility criteria can be empowered, supported and authorized. Balance is the key to preserve the beneficial outcomes for everyone.

Patient partnership attains the patient partner and the frontline staff satisfaction simultaneously. When adopted by healthcare organizations, a single process like partnership can achieve major healthcare quality goals without the need for maximizing efforts. Patient partnership needs management support and staff motivation to embed the process in the quality management culture.

## Conclusion

Partnership is needed to raise awareness among healthcare organizations and the general public. Developing countries can benefit from our experience to assess implementing PFPC in their healthcare facilities. This study showed that the implementation of PFPC in a Lebanese hospital produced favorable outcomes. The main PFPC advantages were patient loyalty preservation, achieving patient satisfaction and quality improvement without limitations. Partnership can be additionally improved by pre-PFPC training, real PP experience sharing, assigning a PFPC coordinator staff and proper time allocation.

The patient and family members recruited in this study were valued as underutilized resources. They have the potential to directly aid a healthcare organization in achieving favorable outcomes of safe and high-quality care. This study highlights the importance of patient empowerment and its ability to bring changes that expert healthcare providers could not achieve alone. Finally, we recommend standardizing the PFPC in the patient and family rights chapter of the Lebanese accreditation.

## Supporting information

**S1 Fig. Gantt chart representation of the timeline of PFPC implementation at SGH.**
(TIF)

**S1 File.**
(DOCX)

## Acknowledgments

This study was the fruit of a collaborative master's program between the Lebanese University and Gates Group. We thank the Saint George Hospital administration and staff for their support in the successful implementation of the PFPC. We also thank the patient partners who participated in this study for their time and effort allocation.

## Author Contributions

**Conceptualization:** Alaa Yehya Dayekh, Mohammad Naseridine, Adel Olleik.

**Data curation:** Alaa Yehya Dayekh.

**Formal analysis:** Alaa Yehya Dayekh.

**Investigation:** Alaa Yehya Dayekh, Mohammad Naseridine.

**Methodology:** Alaa Yehya Dayekh, Mohammad Naseridine.

**Project administration:** Mohammad Naseridine, Adel Olleik.

**Resources:** Fatima Dakroub, Adel Olleik.

**Supervision:** Mohammad Naseridine, Fatima Dakroub.

**Validation:** Fatima Dakroub, Adel Olleik.

**Visualization:** Alaa Yehya Dayekh, Fatima Dakroub.

**Writing – original draft:** Alaa Yehya Dayekh.

**Writing – review & editing:** Alaa Yehya Dayekh, Mohammad Naseridine, Fatima Dakroub, Adel Olleik.

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
