## [Decision Letter · Decision Letter 0]

7 Mar 2022

PONE-D-21-25886Impact, obstacles and boundaries of patient partnership: a qualitative interventional study in LebanonPLOS ONE

Dear Dr. Dayekh,

Thank you for submitting your manuscript to PLOS ONE. After careful consideration, we feel that it has merit but does not fully meet PLOS ONE’s publication criteria as it currently stands. Therefore, we invite you to submit a revised version of the manuscript that addresses the points raised during the review process.

We look forward to receiving your revised manuscript.

Kind regards,

Paavani Atluri

Academic Editor

PLOS ONE

3. PLOS requires an ORCID iD for the corresponding author in Editorial Manager on papers submitted after December 6th, 2016. Please ensure that you have an ORCID iD and that it is validated in Editorial Manager. To do this, go to ‘Update my Information’ (in the upper left-hand corner of the main menu), and click on the Fetch/Validate link next to the ORCID field. This will take you to the ORCID site and allow you to create a new iD or authenticate a pre-existing iD in Editorial Manager. Please see the following video for instructions on linking an ORCID iD to your Editorial Manager account: https://www.youtube.com/watch?v=_xcclfuvtxQ.

Reviewers' comments:

Reviewer's Responses to Questions

**Comments to the Author**

1. Is the manuscript technically sound, and do the data support the conclusions?

Reviewer #1: Partly

Reviewer #2: Partly

2. Has the statistical analysis been performed appropriately and rigorously? 

Reviewer #1: Yes

Reviewer #2: I Don't Know

3. Have the authors made all data underlying the findings in their manuscript fully available?

Reviewer #1: No

Reviewer #2: Yes

4. Is the manuscript presented in an intelligible fashion and written in standard English?

Reviewer #1: No

Reviewer #2: Yes

5. Review Comments to the Author

Reviewer #1: The current study focuses on a patient partnership experience in a Lebanese hospital. This is an interesting study relevant to the current qualitative research in the developing low-middle income African countries such as Lebanon, where human and financial resources are limited. The present study was carried out during this challenging COVID-19 pandemic, where a hospital-based study was difficult to conduct.

There are a few points which may improve the present manuscript are enlisted below:

Specific points

1. The whole manuscript (MS), especially the Introduction, needs to be proofread for coherence and grammar. A few examples are page 4, line 66 use of ‘unsurprisingly’ or use of ‘competent’ on page 10 line 180 was not apt.

2. An extended form and the abbreviation in the first encounter and only the abbreviation later on. Some of the abbreviations, such as QIP, the extended form, was missing in the text. [Page 2, line 30]

3. Author should add the ethical committee approval number in the MS. [Page 6, line no 105]

4. On Page 7, lines 119-120, the author should clearly mention the process of study participants, and the denominator was missing in the text. Among how many patients, x number of patients were eligible and among them how many patients were willing to participate, similarly, for the hospital staff.

5. Reference for SPSS version was missing. [Page 9, Line 177 ]

6. As this is a qualitative study, the author should mention which other software and techniques have been used for the data analysis, such as NVIVO etc.

7. In the result section, Table 2 QIP topic is unclear; what do they represent?

8. Given the small sample size, in Table 2, Four Age categories may not be useful. I can suggest merging it into two categories less than 40 or More than 40. Also, age as a continuous variable with Median (SD) should be added.

Overall, this is an interesting study; however, it requires major English proofreading and minor edits in the MS.

Reviewer #2: Applaud the attempt, patient participation is key in good outcomes. However, the way the inclusion for patients was set up, there is inherent bias. These appear to be patients that are more willing and capable to participate. This does not detract from the findings but can be a launching point for a follow up study to try and recruit patients that may need more help getting involved.

6. PLOS authors have the option to publish the peer review history of their article (what does this mean?). If published, this will include your full peer review and any attached files.

Reviewer #1: No

Reviewer #2: **Yes: **Isha Shah

---

## [Author Response · Author response to Decision Letter 0]

18 Apr 2022

We would like to thank the reviewers for their thorough evaluation and comments that helped to improve our manuscript. For easier visualization, the reviewers’ comments are formatted in the bold and italic font styles. Please note that the line numbers mentioned in this document are compatible with those of the clean version of the manuscript. 

Reviewer 1

The current study focuses on a patient partnership experience in a Lebanese hospital. This is an interesting study relevant to the current qualitative research in the developing low-middle income African countries such as Lebanon, where human and financial resources are limited. The present study was carried out during this challenging COVID-19 pandemic, where a hospital-based study was difficult to conduct. There are a few points which may improve the present manuscript are enlisted below:

Thank you for the positive evaluation and for your comments.

1. The whole manuscript (MS), especially the Introduction, needs to be proofread for coherence and grammar. A few examples are page 4, line 66 use of ‘unsurprisingly’ or use of ‘competent’ on page 10 line 180 was not apt.

The whole manuscript was revised for grammar and coherence.

2. An extended form and the abbreviation in the first encounter and only the abbreviation later on. Some of the abbreviations, such as QIP, the extended form, was missing in the text. [Page 2, line 30]

We added the extended form for each abbreviation upon its first appearance in the text. The abbreviation list was proofread to include all the abbreviations mentioned in the text in alphabetical order. 

3. Author should add the ethical committee approval number in the MS. [Page 6, line no 105]

Thank you for your comment. However, an approval number is not applicable in our case. The Saint Georges Hospital’s ethical committee provides written approval letters for qualifying research applications. We noticed from our letter that they do not seem to include a corresponding approval number in such letters. We contacted the hospital for assistance and it appears that a specific approval number is not usually granted for all applicants to the ethics committee. 

4. On Page 7, lines 119-120, the author should clearly mention the process of study participants, and the denominator was missing in the text. Among how many patients, x number of patients were eligible and among them how many patients were willing to participate, similarly, for the hospital staff.

The sampling method was a purposeful guided approach, where sample selection was based on preset inclusion criteria for both patient partners and healthcare professionals. 

It is important to emphasize on the fact that willingness to participate was one of the inclusion criteria elements. Thus, if a participant met all other elements of eligibility and was not motivated or didn’t have the will to participate, he/she was considered ineligible. 

• The requested denominators were added to the materials and methods section, recruitment and selection sub-section, Lines 120-124:

{Out of 1997 hospitalized patients, we contacted 70 candidates. Only 41 patients and family members were eligible and willing to participate as a patient partner. Out of 389 healthcare employees, 121 were multidisciplinary healthcare workers with a minimum of two years of experience and a previous quality improvement expertise. However, only 27 HP were willing to participate and thus considered eligible and included.}

5. Reference for SPSS version was missing. [Page 9, Line 177]

• The SPSS version reference was added to the materials and methods section, data analysis sub-section, Lines 185-186: 

‎{The coded data were entered and analyzed using the IBM SPSS statistics software (IBM SPSS Statistics for Windows, Version 25.0. Armonk, NY: IBM Corp.)}

6. As this is a qualitative study, the author should mention which other software and techniques have been used for the data analysis, such as NVIVO etc.

We didn’t use other software for qualitative data analysis. The research team used paper coding for both the open and selective coding phases. 

7. In the result section, Table 2 QIP topic is unclear; what do they represent?

Thank you for this comment. We agree that this data needs clarification and have addressed the issue in the manuscript.

One of the study’s goals was to implement a quality improvement process (QIP) through the patient and family partnership committee (PFPC) meetings. For the purpose, the participants had to choose one priority focus area to analyze its root causes, discuss the opportunities for improvement and suggest corrective actions. 

Three topics were suggested by the research team: conversation time with patients (i), work place violence (ii), and hand hygiene compliance (iii). These priority focus areas are actually common quality measures that are sustainably monitored by the healthcare quality staff and policy makers in the healthcare field. 

All 68 participants (27 HP and 41 PP) in the PFPC chose to discuss the ‘‘conversation time with patients’’ quality measure. 

In summary, healthcare practitioners are not able to provide sufficient time for a meaningful conversation with patients when providing care. As a result, patients and their families may lack the important health education needed for an effective self-management of their illnesses. Both patients and healthcare practitioners had realized the hazardous effects of conversation time decrease during healthcare service delivery. They eventually agreed to enhance this process for a safer care through partnership.

Introduced changes:

• To clarify the idea for the reader, a footnote was added to Table 2 listing the QIP topic choices. Moreover, a short abbreviation list was added beneath the table to demonstrate the extended form of QIP.

• Information in the text can be found in the materials and methods section, methodology sub-section, Lines 147-148:

{ The participants selected a QIP topic to discuss in the PFPC among three options: conversation time with patients (i), workplace violence (ii) and hand hygiene compliance (iii). }

8. Given the small sample size, in Table 2, Four Age categories may not be useful. I can suggest merging it into two categories less than 40 or More than 40. Also, age as a continuous variable with Median (SD) should be added.

We agree with the reviewer and thank them for the insightful comment. 

• The corresponding changes are shown in Table 2:

Age was categorized into two categories: less than 40 years (i) and more than 40 years (ii). 

A row was added to represent age as a median with standard deviation (SD).

Overall, this is an interesting study; however, it requires major English proofreading and minor edits in the MS.

Thank you for your comments which helped to increase the manuscript’s quality. We hope that the major English proof-reading request and the minor manuscript edits were addressed successfully.

Reviewer 2

Reviewer #2: Applaud the attempt, patient participation is key in good outcomes. 

Thank you for the constructive feedback and encouragement.

We support your notion that patient participation is key in good outcomes. This study is a light of hope in the journey to the implement patient and family partnership as the most advanced model of care in a low-income country. Patient partnership emphasizes on the crucial empowerment and authorization of the patient and his experiential knowledge. It engages patients in all aspects of healthcare system design, delivery and monitoring, rather than just the treatment plan. 

However, the way the inclusion for patients was set up, there is inherent bias. These appear to be patients that are more willing and capable to participate. This does not detract from the findings but can be a launching point for a follow up study to try and recruit patients that may need more help getting involved.

We agree about the bias that can be introduced by the selection criteria. One of the major barriers that we faced during implementation was the healthcare practitioners’ presumption of the incapability of patients to decide (due to their lack of academic knowledge and expertise in healthcare). Thus, if we have not selected partners with specific characteristics, this new phenomenon would have faced major barriers in its implementation for a first time in a healthcare setting. 

Moreover, this was a preliminary investigation to explore the patient partnership phenomenon and its specific benefits and barriers in an under-resourced developing country. The identified elements of the partnership experience will be indispensable for effective and reliable future investigations. After this successful implementation, we are currently preparing for a follow-up study that will widen the aspect of participation from different patient groups, including patients with barriers and difficulties. We hope that by the end of our investigation, Lebanese hospitals will be encouraged to consider patient partnership in their quality improvement approaches.

• We reported the bias in the sampling method in the materials and methods section, data analysis sub-section, Lines 188-189:

{The research team focused on identifying and minimizing potential sources of bias. The sampling method may constitute a potential source of bias by only recruiting patients who were more capable or willing to participate.}

---

## [Decision Letter · Decision Letter 1]

15 Jun 2022

Impact, obstacles and boundaries of patient partnership: a qualitative interventional study in Lebanon

PONE-D-21-25886R1

Dear Dr. Dayekh,

We’re pleased to inform you that your manuscript has been judged scientifically suitable for publication and will be formally accepted for publication once it meets all outstanding technical requirements.

Kind regards,

Paavani Atluri

Academic Editor

PLOS ONE

Additional Editor Comments (optional):

Reviewers' comments:

Reviewer's Responses to Questions

**Comments to the Author**

1. If the authors have adequately addressed your comments raised in a previous round of review and you feel that this manuscript is now acceptable for publication, you may indicate that here to bypass the “Comments to the Author” section, enter your conflict of interest statement in the “Confidential to Editor” section, and submit your "Accept" recommendation.

Reviewer #1: All comments have been addressed

2. Is the manuscript technically sound, and do the data support the conclusions?

Reviewer #1: Yes

3. Has the statistical analysis been performed appropriately and rigorously? 

Reviewer #1: Yes

4. Have the authors made all data underlying the findings in their manuscript fully available?

Reviewer #1: Yes

5. Is the manuscript presented in an intelligible fashion and written in standard English?

Reviewer #1: Yes

6. Review Comments to the Author

Reviewer #1: Authors have successfully addressed all the major and minor comments. Best wishes to the authors for their further research. I have no further concerns.

7. PLOS authors have the option to publish the peer review history of their article (what does this mean?). If published, this will include your full peer review and any attached files.

Reviewer #1: **Yes: **Prabal Chourasia

---

## [Editor Report · Acceptance letter]

28 Jun 2022

PONE-D-21-25886R1 

Impact, obstacles and boundaries of patient partnership: a qualitative interventional study in Lebanon 

Dear Dr. Dayekh:

I'm pleased to inform you that your manuscript has been deemed suitable for publication in PLOS ONE. Congratulations! Your manuscript is now with our production department. 

Kind regards, 

on behalf of

Dr. Paavani Atluri 

Academic Editor

PLOS ONE